# PhysicalTwin: Mixed Reality Interaction Environment for AI-supported Assistive Robots

**Max Pascher**
max.pascher@w-hs.de
Westphalian University of Applied
Sciences
Gelsenkirchen, Germany
University of Duisburg-Essen
Essen, Germany

**Kirill Kronhardt**
kirill.kronhardt@w-hs.de
Westphalian University of Applied
Sciences
Gelsenkirchen, Germany

**Jens Gerken**
jens.gerken@w-hs.de
Westphalian University of Applied
Sciences
Gelsenkirchen, Germany

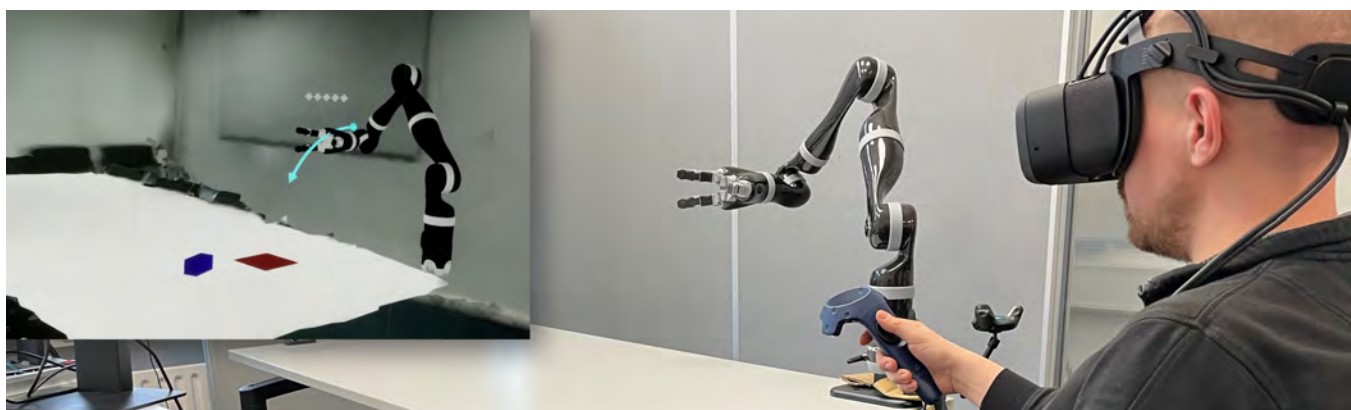

**Figure 1: Mixed Reality setup with (a) the user's view in simulation and (b) an over-the-shoulder image from reality**

## ABSTRACT

Robotic arms belong to a group of innovative and powerful assistive technologies which can support people with motor impairments. These assistive devices allow people to perform Activities of Daily Living (ADLs) involving grasping and manipulating objects in their environment without caregivers' support. However, controlling a robotic arm can be a challenging and time-consuming task. Further, assistive robot technologies' often bulky, expensive, and complex nature makes developing and testing new interaction and control options a laborious and potentially precarious endeavor. To address this, we designed and developed *PhysicalTwin* - a Mixed Reality (MR) interaction environment for AI-supported assistive robots. *PhysicalTwin* contains a virtual *Kinova Jaco* assistive robotic arm, which can be controlled via different AI-supported control methods. The virtual *Kinova Jaco* can also be mirrored directly via Robot Operating System (ROS) to its real-world version for direct control. The entire MR continuum can be used to interact with the robot and receive visual feedback, resulting in a fully customizable environment for the user.

## CCS CONCEPTS

• **Computer systems organization** → **Robotic control**; • **Human-centered computing** → *Visualization techniques*; *Virtual reality*.

*VAM-HRI '23, March 13, 2023, Stockholm, Sweden*
2023. ACM ISBN 978-x-xxxx-xxxx-x/YY/MM…$15.00
https://doi.org/10.1145/nnnnnnn.nnnnnnn

## KEYWORDS

assistive robotics, human-robot interaction, shared user control, mixed reality, visual cues

**ACM Reference Format:**
Max Pascher, Kirill Kronhardt, and Jens Gerken. 2023. PhysicalTwin: Mixed Reality Interaction Environment for AI-supported Assistive Robots. In *VAM-HRI '23: A6th International Workshop on Virtual, Augmented, and Mixed-Reality for Human-Robot Interactions at HRI 2023, March 13, 2023, Stockholm, Sweden.* ACM, New York, NY, USA, 5 pages. https://doi.org/10.1145/nnnnnnn.nnnnnnn

## 1 INTRODUCTION

According to a recent World Health Organization report [21], 16% of the global population has some form of disability. Among those affected, many have compromised mobility that limits their ability to care for themselves without assistance from others, leading to a constant need for caregivers [12]. Investigative research by Pascher et al. highlighted that people with physical impairments often desire privacy and alone-time, which could be facilitated by reliable robotic support [14]. Additionally, Kyrarini et al. illustrated in their comprehensive review the positive impact of assistive robotic systems – known as cobots – in supporting people with motor impairments [10] in Activities of Daily Living (ADLs). This allows people who were previously reliant on others to regain their independence by reducing the constant presence of caregivers.

However, the use of (semi-) autonomous robotic systems can be challenging and lead to additional stress for end-users if not correctly addressed during the design process [17]. Particularly close proximity collaboration between humans and robots often remains challenging [6]. These challenges include effective communication to the end-user of (a) motion intent [15] and (b) the spatial perception of the robot's vicinity [16]. The decreased feeling of control experienced by users in autonomous mode has been convincingly demonstrated in a study by Pollak et al.. They found that switching to manual mode allowed participants to regain control and significantly decrease stress [17]. These findings are consistent with a comparative study by Kim et al., again highlighting that manual mode is associated with higher user satisfaction [8].

In care environments, which focus on fulfilling flexible demands, cobots assist their users in diverse ways, leading to challenges in handling robots safely and effectively [4]. The type of robot used usually has multiple Degrees-of-Freedom (DoFs), requiring complex input devices or a division into different modes with joystick controls [11, 18]. These commonly used control methods are frequently unsuitable for people with motor impairments [13].

Adaptive controls are a potential solution to these challenges as they merge the advantages of (semi-) autonomous actions with the flexibility of manual controls [5]. They dynamically combine the DoFs of the robot for a specific scenario to assist in its control. However, developing and testing new interaction and control concepts can often be limited due to the physical bulk and complexity of the real robot. This is, in particular, true when such interaction concepts go beyond the robot, by changing the physical setup, adding feedback components or even providing information to the user in Augmented Reality (AR).

As a way to more easily and safely test such novel interaction concepts, we designed *PhysicalTwin* - a Mixed Reality (MR) interaction environment for AI-supported assistive robots. *PhysicalTwin* also allows researchers to rapidly design and test novel visualization and interaction methods, without requiring additional hardware or interfaces between devices to be developed. In addition, the virtual *Kinova Jaco* can be mirrored directly via Robot Operating System (ROS) to its real world version for direct control. As the entire MR continuum can be used, integration of novel interaction designs and feedback techniques to interact with robotic arms into the actual routines of the end-user becomes easier.

## 2 SIMULATION ENVIRONMENT

Our testbed environment – *PhysicalTwin* – contains a virtual *Kinova Jaco*[1] assistive robotic arm and includes different control mechanisms as well as visual feedback mimicking AR. We designed the Virtual Reality (VR) environment based on a photogrammetry scan of a blank room. Inside this virtual room, a simulated model of the *Kinova Jaco* is attached to a table (see Figure 2). Using a virtual model of a real robotic arm allowed us to stay as close to an actual physical system as possible. Additionally, the *Kinova Jaco* is specifically designed and commonly used as an assistive device for people with motor impairments [1, 7]. As most real-world scenarios will include pick-and-place operations, we designed a straightforward

testbed scenario for these tasks. The main elements are a red target surface as a drop target to place an object and a blue block as the picking object. The task is designed to provide a randomized positioning of the blue block, which will reappear once a placement on the red surface has been successful. In order to support more formal studies and experiments, two virtual screens have been added – one for descriptions and questionnaires and one capable of showing example images of the control types.

The virtual environment was created with the *Unreal Engine 4.26* and optimized for the *Meta Quest2* VR headset. As the simulation runs as a standalone app, it can be set up on comparable Head-Mounted Display (HMD) with minimal effort. The *Quest* motion controller is used in the simulation to operate the robotic arm. The associated analog stick and control buttons provide an excellent foundation to implement and test various interaction concepts, including adaptive concepts which can be configured to match the individual physical abilities of the user.

As a simplification of the robot behavior in 3D space, the user in the simulation moves and operates the gripper, and the robot arm is programmed to adopt a correct pose automatically. This was implemented using the physics system of the *Unreal Engine*.

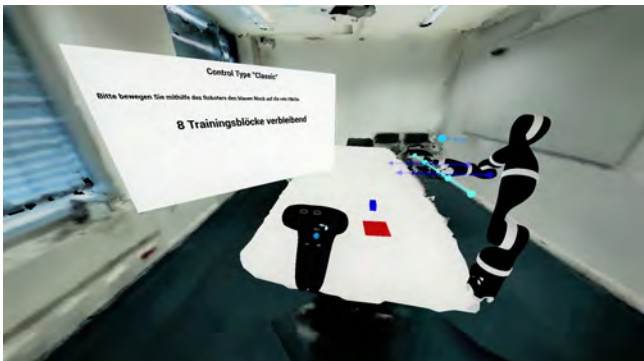

**Figure 2: Virtual environment consisting of (left to right): a virtual canvas, the motion controllers, a table with a blue object and red target, and a *Kinova JACO* with an arrow-based visualization.**

## 3 AI-SUPPORTED CONTROL AND VISUAL CUES

Controlling robotic arms can be difficult and time-consuming, especially for novice or non-tech-savvy users [7]. People with motor impairments are often limited in their ability to control complex input devices [13]. Studies indicate that they mostly rely on moving a joystick back and forth in one direction, corresponding to one DoF or similar limited input devices (e.g., head movements [19]). Operating a robotic arm in 3D space will thus require a constant change in mapping this single DoF. A likely scenario is that moving the joystick back and forth is mapped to moving the robotic arm forward and backward. However, to move the device up and down, a mode switch is necessary to change this mapping and allow the same joystick movement to operate the robotic arm in that direction. These devices require up to 7 DoF to be operated in 3D space,

---

[1]Kinova Jaco robotic arm: https://assistive.kinovarobotics.com/product/jaco-robotic-arm, last retrieved March 13, 2023

including opening and closing the gripper. Suitable input formats (e.g., joysticks) mostly have two DoFs or add significant interaction complexity to include more DoFs.

Our research investigates the possibility of decreasing the difficulty of controlling a high-DoF assistive robotic arm using a low-DoF input device. Adaptive controls allow for the dynamic and semi-automatically assignment of different combinations of movement-DoFs to the device's input DoFs depending on the current situation. On a technical level, this is achieved through an AI system – a Convolutional Neural Network (CNN) – which generates and suggests DoF mappings based on real-time camera data of the current situation. The CNN is trained on a dataset of 10.000 motions of pick-and-place tasks. The result is a set of updated mapped DoFs, ranked by their assumed usefulness for the given situation, allowing users to access a variety of movements for each scenario. This approach is an improvement over previous work as it is not limited to cardinal DoFs or pre-determined motions of an autonomous system, whilst also being able to represent those.

One potential downside of this approach is the decreased legibility of the robotic arm behavior, i.e., the user might have difficulties understanding what kind of robot movement a particular input device interaction, such as moving a joystick up, will actuate. To improve the predictability of the system and thus improve user acceptance, it is necessary to communicate the intended DoF mapping provided by the autonomous system [2]. Information about a robot's plan and activity can help users better understand and expect a robot's behavior [20].This is in line with research by Cleaver et al. who showed that users generally prefer to have the robot's future movements represented visually [3].

The problem of communicating robot motion intent has been studied extensively with different types of visualization and modalities used [15, 20]. Our current research focus aims to overcome this by providing augmented visual cues to communicate robot motion intent. The newly designed *PhysicalTwin* testbed environment allows for easy implementation and testing of different augmented visualization concepts. Additionally, we have developed various visualizations, which enable the user to see and understand how the input control will be mapped to the robot arm movement (see Figure 3). Users will therefore be able to switch between AI-suggestions or use a non-AI-supported control type.

**Ghost:** A visualization of robot motion intent by showing an additional version of the robot (or specific components) registered in 3D space, in another color and/or opacity. These visualizations communicate the exact position and orientation a robot at a given time, behaving precisely as though the real robot had been moved this way.

**Waypoints:** This visualization technique augments the position of a robot (or in our case, the gripper of the robotic arm) in 3D space at a certain point in the future. Usually, the robot navigates linearly between these *Waypoints*, which increases predictability.

**Arrow:** Among visualizations arguably the most basic but certainly also the most familiar (as seen in traffic navigation systems, road signs, and on keyboards). *Arrows* are found both in straight and curved varieties, where curved arrows indicate a rotation. Given the abundance of *Arrows* in daily life, it makes sense that many robot motion intent visualizations use them.

**Classic:** This visualization also uses *Arrows*, but in our prototype they are used as a baseline condition to evaluate adaptive and non-adaptive controls. Here, as with the standard input device *Kinova Jaco*, two axes can be controlled simultaneously and the user has to choose between different translations and rotations by mode-switching.

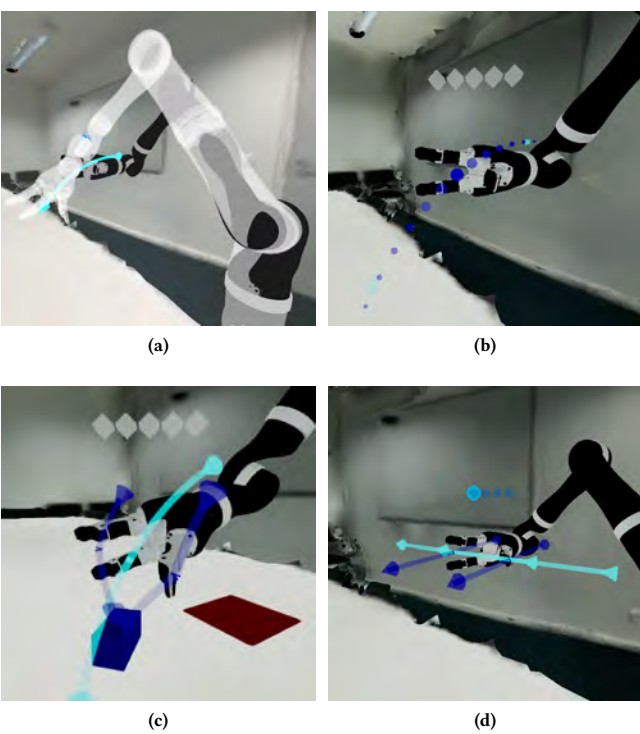

**(a)**      **(b)**

**(c)**      **(d)**

**Figure 3: Visualization examples for an AI-supported robotic control: (a) Ghost, (b) Waypoints, (c) Arrow, and non AI-supported: (d) Classic.**

## 4 USING PHYSICALTWIN FOR REMOTE STUDIES

With *PhysicalTwin*, researchers can distribute the study environment to participants who own a VR-headset like the *Meta Quest* or *Meta Quest2* or visit participants in their homes to perform the study. The innovative advantage is that the actual physical robotic device need not be present at all. This significantly increases the potential to include end-users in the research and design process while at the same time decreasing cost and time involvement.

As a proof of concept, we ran an unmonitored remote study with *PhysicalTwin* and our arrow-based AI-supported control mechanism with 39 participants [9]. Here, we compared two adaptive control methods with the standard mode-switch control type *Classic*, explicitly focusing on task completion times, number of mode switches and workload. In addition, we received qualitative results from voice recordings of our participants, providing a deeper understanding of the benefits and challenges of each of the three employed control types. During the study, participants controlled

the robot using the right motion controller of the VR headset. In particular, the control stick of the motion controller moved the robot according to the currently active control type (either a *Classic* control type or one of two AI-supported control types). This enabled the participants to control which DoFs were being used and how fast the robot would move. The A-Button was used to switch to the next mode cyclically, returning to the first mode when a mode switch was performed in the last mode. Results show that the number of mode switches necessary to complete a simple pick-and-place task decreases significantly when using an adaptive, AI-supported control method. Based on this, we predict the success of using *PhysicalTwin* in remote studies to evaluate interaction designs and feedback techniques in Human-Robot Interaction (HRI).

## 5 BIDIRECTIONAL ROS INTEGRATION

The ROS integration allows for a bidirectional exchange of information between the framework and a real robot, mirroring the robot's state in the simulation or vice versa. Figure 4 shows the involved components.

A ROS bridge establishes a multi-device connection between the framework and the real robot while exchanging robot data between both two. On the ROS side, the messages for controlling the arm in position and orientation as well as for the values for the angle-accurate control of the gripper fingers are subscribed in our ROS node. After extracting and pre-processing of the values, the robotic arm and gripper are controlled by our ROS action client. In addition, the joint angles, the Tool Center Point (TCP) and the position of all three gripper fingers are published via ROS, which are then input by our *Unreal Engine* framework. The virtual and real robot are synchronized via ROS every 0.1 seconds.

Based on this, our framework provides – depending on the specific context – both a *DigitalTwin* and *PhysicalTwin* approach, allowing the control of either with the other.

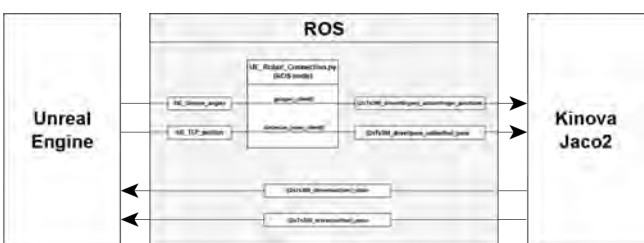

**Figure 4: Communication scheme of the ROS integration for mirroring simulation and real robot.**

## 6 VISUALIZING IN MIXED REALITY

Intending also to be able to control and interact with robotic arms in the non-virtual world via our ROS integration, we have developed *PhysicalTwin* for use on the *Varjo XR-3* MR-headset. The headset works with two high-resolution stereoscopic cameras, which allow the user to take advantage of the entire continuum of MR. By using two *VIVE* trackers, the virtual and real world are synchronized so that the robot's working areas are identical in both worlds. With the help of the *VIVE* motion controller, it is then possible to control the robot directly via *PhysicalTwin* (see Figure 5).

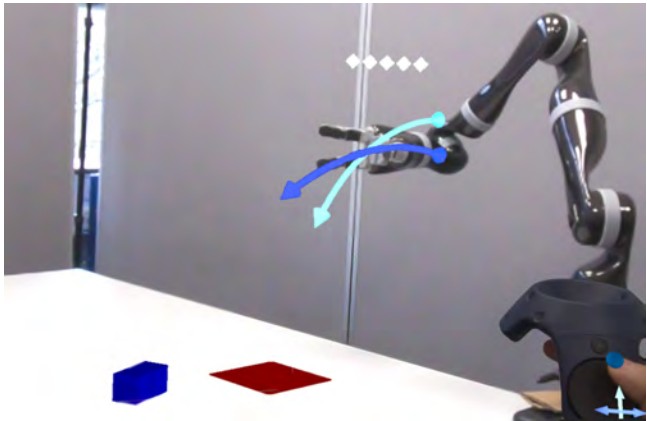

**Figure 5: Mixed Reality view of the *Kinova Jaco* robotic arm. The cyan arrow indicates the current movement direction of the arm, while the blue arrow shows the recommendation of the AI-supported control, which can be accepted by the user to change DoF-mapping and therefore movement direction.**

The virtual and real environment of the robotic arm are perfectly aligned, allowing us to seamlessly switch between the user controlling the real and virtual robot. The level of MR can be adjusted in various steps. These setups of the MR environment include:

(1) the completely real environment with the real robot arm,
(2) the real environment extended with visual cues,
(3) the real environment in which the virtual robot is transferred and displayed (with and without visual cues),
(4) the virtual environment in which the real robot is transferred and displayed (with and without visual cues),
(5) the completely virtual environment with the virtual robot arm.

## 7 CONCLUSION

Our work on *PhysicalTwin* has demonstrated its effectiveness and usefulness for research and development of assistive robotic arms. Still, there are certain limitations on the fidelity and immersion in such an environment compared to interacting with a real robot. However, we believe that the vastly reduced overhead of a complex physical setup of a real robot in the early phases of ideation and prototyping far outweigh these constraints. Furthermore, end-users' involvement can be increased, as *PhysicalTwin* neither requires moving a physical robot to a different place (e.g., visiting a participant) nor is the approach restricted to lab studies.

The integration of ROS extends *PhysicalTwin* to be an MR solution for controlling a robotic arm. Using the full continuum of MR allows researchers to develop and evaluate new feedback methods and visualizations. These enable users to use AI-supported control to easily operate a multi-DoF robotic arm in real-world. As MR technologies continue to evolve, we can expect these devices to become more powerful, smaller and more efficient. This is a prerequisite for everyday use and easier integration into the actual routines of the end-user. Further development of MR technologies enable novel interaction designs and feedback techniques to interact with robotic arms, for example, supporting people in their ADLs at home.

## ACKNOWLEDGMENTS

We thank our students Maximiliano Adaro, Christine Schreiber, Darius Antoni, and Kevin Zinta for their dedication and valuable assistance during the project.

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
