# OpenReview forum: "PhysicalTwin: Mixed Reality Interaction Environment for AI-supported Assistive Robots"
_humanrobotinteraction.org/HRI/2023/Workshop/VAM-HRI — VAM-HRI 2023 Oral_

### Official Review · Program_Chairs · 2023-02-25
**Accept**

**Rating:** 7
**Confidence:** 5

**Review:**

Review 1:

This paper describes PhysicalTwin, a mixed reality testbed for human operation of assistive robotic arms. Initially implemented in VR, PhysicalTwin provides a controllable model of a Kinova Jaco arm, inserted into a mimicry of a real world environment generated by photogrammetry data. In this system, AR-style visualizations are able to be overlaid onto the robot or environment and can be used to design fully virtualized user studies regarding effective teleoperation or interaction with assistive robotic arms, without requiring a physical robot to be present. PhysicalTwin has a built-in communication schema using ROS, allowing it to also be ported to AR, overlaying the same visualizations on a real-world robot.

Strengths:
- PhysicalTwin seems like a useful tool for conducting research into assistive HRI, allowing for rapid prototyping of interaction modes and visualizations, and potentially allowing for large-scale data collection without needing to physically bring participants in contact with a robot.
- PhysicalTwin possesses a modular design, allowing its programmed visualizations to easily be transferred to a real-world robotic interaction environment.
- The concept of crowdsourcing unmonitored experiment data remotely for people who possess a VR headset is something that I’ve never seen before in HRI research, and seems like an interesting and powerful way of reaching target populations who may be in short supply around a university campus.

Weaknesses/Questions:
- I know the user study undertaken with PhysicalTwin is described in more detail in another paper, but since a key part of this paper’s intended contribution is demonstrating how this software enables user studies to be effectively run in an unmonitored, decentralized manner, it could use a bit more detail in the paper. At the moment, section 4 seems too abstract to get a sense of how the study was actually conducted/what its hypotheses and findings were.
- I think, seeing as this paper is describing a software tool, a bit more discussion on future work/potential applications of the technology could be included.
- Again, seeing as this is a paper about a certain system, it would be a bonus to include some link to the software (even if it’s a link to the app downloaded as part of the user study). That would increase the paper’s usefulness to its future readers.

In summary, I think this paper is a great fit for VAM-HRI, and I recommend acceptance.

Reviewer 2:
This paper presents PhysicalTwin, an MR interface for assistive teleoperation. Overall I recommend this paper be accepted, it is relevant to the community, and addresses an interesting problem (controlling a high DoF arm in 3d space with VR).

Feedback:
- More details on the CNN used and training process would be good to include, such as the architecture details, training / evaluation loss, etc.
- More details on the different visualization examples shown in Figure 3 would be helpful.

---

### Decision · Program_Chairs · 2023-03-02

Accept (Oral)